# QUERY THE AGENT: IMPROVING SAMPLE EFFICIENCY THROUGH EPISTEMIC UNCERTAINTY ESTIMATION

## ABSTRACT

Curricula for goal-conditioned reinforcement learning agents typically rely on poor estimates of the agent's epistemic uncertainty or fail to consider the agents' epistemic uncertainty altogether, resulting in poor sample efficiency. We propose a novel algorithm, Query The Agent (QTA), which significantly improves sample efficiency by estimating the agent's epistemic uncertainty throughout the state space and setting goals in highly uncertain areas. Encouraging the agent to collect data in highly uncertain states allows the agent to improve its estimation of the value function rapidly. QTA utilizes a novel technique for estimating epistemic uncertainty, Predictive Uncertainty Networks (PUN), to allow QTA to assess the agent's uncertainty in all previously observed states. We demonstrate that QTA offers decisive sample efficiency improvements over preexisting methods.

## 1 INTRODUCTION

Deep reinforcement learning has been demonstrated to be highly effective in a diverse array of sequential decision-making tasks (Silver et al., 2016; Berner et al., 2019). However, deep reinforcement learning remains challenging to implement in the real world, in part because of the massive amount of data required for training. This challenge is acute in robotics (Sünderhauf et al., 2018; Dulac-Arnold et al., 2019), in tasks such as manipulation (Liu et al.), and in self-driving cars (Kothari et al.).

Existing curriculum methods for training goal-conditioned reinforcement learning (RL) agents suffer from poor sample efficiency (Dulac-Arnold et al., 2019) and often fail to consider agents' specific deficiencies and epistemic uncertainty when selecting goals. Instead, they rely on poor proxy estimates of epistemic uncertainty or high-level statistics from rollouts, such as the task success rate. Without customizing learning according to agents' epistemic uncertainties, existing methods inhibit the agent's learning with three modes of failure. Firstly, a given curriculum may not be sufficiently challenging for an agent, thus using timesteps inefficiently. Secondly, a given curriculum may be too challenging to an agent, causing the agent to learn more slowly than it could otherwise. Thirdly, a curriculum may fail to take into account an agent catastrophically forgetting the value manifold in a previously learned region of the state space. Curriculum algorithms need a detailed estimate of the agent's epistemic uncertainty throughout the state space in order to maximize learning by encouraging agents to explore the regions of the state space the agent least understands (Kaelbling, 1993; Plappert et al., 2018).

We propose a novel curriculum algorithm, Query The Agent (QTA), to accelerate learning in goal-conditioned settings. QTA estimates the agent's epistemic uncertainty in all previously observed states, then drives the agent to reduce its epistemic uncertainty as quickly as possible by setting goals in states with high epistemic uncertainty. QTA estimates epistemic uncertainty using a novel neural architecture, Predictive Uncertainty Networks (PUN). By taking into account the agent's epistemic uncertainty throughout the state space, QTA aims to explore neither too quickly nor too slowly, and revisit previously explored states when catastrophic forgetting occurs. We demonstrate in a 2D continuous maze environment that QTA is significantly more sample efficient than preexisting methods. We also provide a detailed analysis of how QTA's approximation of the optimal value manifold evolves over time, demonstrating that QTA's learning dynamics are meaningfully driven by epistemic uncertainty estimation. An overview of QTA and our maze environments are shown in 1.

We further demonstrate the importance of utilizing the agent's epistemic uncertainty by extending QTA with a modified Prioritized Experience Replay (Schaul et al., 2016) (PER) buffer. This modified

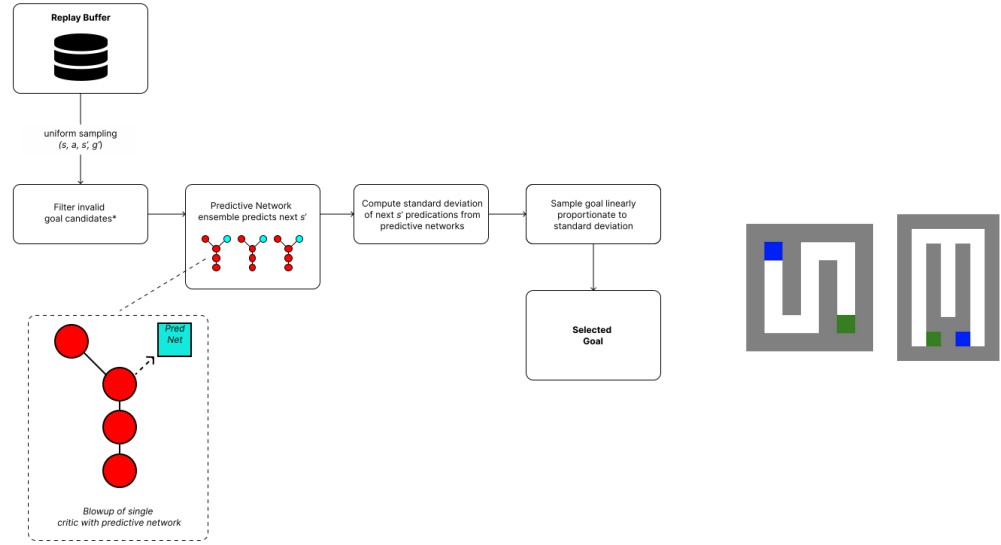

Figure 1: Left: An overview of our goal-selection method. * Threshold filter is only applied when sampling the first goal of an episode. Right: An example of a 1-period square-wave maze, and an M-maze. Starting locations are sampled uniformly in the blue regions and goal locations are sampled uniformly in the green regions. Environment described in detail in section 4

PER buffer assigns sampling priorities to transitions based on states' estimated epistemic uncertainty. QTA augmented with our modified PER buffer outperforms QTA, while QTA using a standard PER buffer does not. This demonstrates the benefits of integrating detailed epistemic uncertainty estimation into reinforcement learning curricula.

Our contributions are:

1. Query the Agent (QTA): A novel curriculum algorithm that adapts goals to the agent's epistemic uncertainty and pushes an agent to improve rapidly by collecting data in highly uncertainty states in the environment.

2. Predictive Uncertainty Networks (PUN): A new technique, broadly applicable to all Q learning approaches, for measuring an agent's epistemic uncertainty throughout the state space by using the agent's own latent representation. We demonstrate an implementation with DDPG.

3. An analysis, including ablation experiments, on how QTA estimates epistemic uncertainty throughout state space and how QTA evolves an agent's understanding of its environment over time.

## 2 RELATED WORK

Recent advances build upon Hindsight Experience Replay (HER) (Andrychowicz et al., 2017) by designing curricula that allow the agent to cleverly select goals to accelerate learning. None of these appropriately take into account the agent's epistemic uncertainty. We will categorize methods according to which information about the agent they utilize to select goals, then discuss them.

Pitis et al. (2020) does not directly take into account any information regarding the agent's epistemic uncertainty or performance. Their method simply fits a kernel density estimate over the set of previously visited goals, then draws samples as goal candidates and selects the sample with the lowest likelihood under the kernel density estimate. This method benefits from reliably producing

increasingly difficult goals for the agent and being immune to problems such as a goal generator network destabilizing. There are multiple drawbacks. Firstly, if an agent suffers from catastrophic forgetting, the curriculum won't be able to correct the problem effectively. Secondly, if the incremental goals are insufficiently/overly challenging for the agent, the curriculum won't properly adapt. Pitis et al. (2020) shares some commonalities with pseudocount techniques in setting goals exclusively based on visitation frequency.

Similarly, count-based and pseudocount-based algorithms (Tang et al.; Bellemare et al.) rely on counting the number of times that a state or a region of state space has been visited. In high dimensional or continuous environments, these algorithms rely on various compression or learned similarity metrics (Machado et al.; Schmidhuber, 2008) or basic statistics in order to identify which regions of the state space are considered similar and adjacent to one another. Intrinsic exploration bonuses can be awarded and goals can be set based on visitation frequencies to accelerate exploration. These methods, taking into account exclusively visitation frequency, do not directly to into account the agent's epistemic uncertainty or performance and may set goals too ambitiously or lazily and won't account for catastrophic forgetting.

Other techniques, such as Bharadhwaj et al. (2020); Portelas et al. (2019); Florensa et al. (2018), only indirectly take into account the agent's understanding of the state space through approximations based on statistics regarding agents' historical ability to reach goals. Since such metrics offer minimal insight into the agent's understanding of the task, these methods select goals that may not be maximally informative to the agent. The major weakness of these methods is that the goal-generation networks can destabilize, and are required to set goals with only very abstract data about the agent's understanding of the state space.

Another class of algorithms, such as Pathak et al.; Zhang et al. (2020), rely on an external ensemble of networks and estimate epistemic uncertainty throughout the state space. Pathak et al. maintains an external network of forward models, which observe the same data as the agent, and uses the disagreement among the forward models as an epistemic uncertainty estimate. Zhang et al. (2020) does the same, but with three external critics. The major problem with these algorithms is that they are estimating the epistemic uncertainty of *other* networks, not the agent's network. Because these external networks' understanding of the state space will drift away from the agent's over time, the epistemic uncertainty estimate of the external networks is not representative of the agent's epistemic uncertainty estimate. Additionally, Zhang et al. (2020) requires privileged access to the entire state space in order to stabilize.

Other curiosity-related methods, such as random network distillation (Burda et al.), use different formulations of a similar mechanism; estimating the uncertainty of networks disconnected from the agent's computation graph. Random network distillation relies on using the agent's experiences to train an external network to predict the output of a frozen, randomly initialized network known as the target network. As an agent visits a particular state more, the external network improves its prediction of the output of the target network, making it an effective exploration technique to provide intrinsic reward bonuses proportionate to the size of the external network's prediction error. However, this approach relies on estimating the agent's error in the random network prediction task to gauge uncertainty in the Q-learning task. These are two very different tasks with different difficulties and will learn at different speeds, thus this technique provides a poor estimate of epistemic uncertainty.

A number of techniques (Chane-Sane et al., 2021; Nachum et al., 2018; Christen et al., 2021; OpenAI et al., 2021) have used multiple reinforcement learning agents in order to solve long-horizon tasks, by either using the agents in an adversarial fashion or in a hierarchy. Using multiple reinforcement learning agents incurs a significant computation expense, and comes along with significant technical challenges such as nonstationarity between agents. Consideration of these techniques is outside the scope of this work, as this work focuses on maximizing the learning of a single agent.

## 3   METHODS

When learning to navigate a goal-conditioned reinforcement learning environment, we seek to collect as much useful information as possible with each rollout, based on which areas of the state space the agent is least certain about. In each episode, QTA measures the agent's uncertainty and samples a goal in a highly uncertain region of the state space in order to gather state transitions that will be

maximally informative to the agent when performing gradient updates. We can ensure that the most important transitions are sampled immediately for gradient updates through a modified version of prioritized experience replay (PER), in which the priorities are determined by the estimated epistemic uncertainty of a state in the replay buffer.

In this section, we first introduce Predictive Uncertainty Networks (PUN) for predicting epistemic uncertainty. Next, we introduce QTA's mechanism for using epistemic uncertainty estimates to select a goal. Lastly, we introduce an extension of QTA in which our customized PER buffer allows for further sample efficiency improvements.

### 3.1 Uncertainty Estimation

**Predictive Uncertainty Networks (PUN)**  PUN utilizes a critic ensemble to enable effective epistemic uncertainty estimation. PUN uses $G$ critics $\phi_{i \in \{1,...,G\}}$, $G$ target critics $\phi_{targ, i \in \{1,...,G\}}$, and in this implementations a standard policy $\pi_\theta$. In our PUN implementation we fix $G = 3$. Having multiple critics providing estimates is essential for assessing the epistemic uncertainty of the network. In order to ensure the stability of an ensemble of critics, we train them with a joint loss function. Given a mini-batch $B = \{(s, a, r, s', g, g')\}$, inspired by Chen et al. (2021) we sample a set of two indices $M$ from $\{1, ..., G\}$ then compute the target:

$$y_t = r_t + \gamma(\min_{i \in M} Q_{\phi_{targ,i}}(s', a', g')), \quad a' \sim \pi_\theta(\cdot | s')$$

Then update $\phi_i$ with gradient descent using

$$\nabla_\phi \frac{1}{|B|} \sum_{(s,a,g) \in B} (Q_{\phi_i}(s, a, g) - y)^2$$

We update target networks using polyak averaging coefficient $\rho$:

$$\phi_{targ,i} \leftarrow \rho \phi_{targ,i} + (1 - \rho)\phi_i$$

Lastly, the policy is updated as in DDPG (Lillicrap et al., 2016).

**Predictive Network**  We assign a predictive network $P_i$ to each critic network, where each predictive network is a single feed-forward layer that takes as input the final latent state of the critic and predicts the next hidden state the agent will observe. More specifically $Q_{backbone,i}(s, a, g)$ yields a final hidden state $\xi_i$, and the final layer of $Q_i$ maps $Q_{i,final}(\xi_i) \to Q_i(s, a, g)$ while the predictive network maps $P_i(\xi_i) \to \tilde{s}'$. We train the predictive network by minimizing the objective:

$$L = \frac{1}{|B|} \sum_{s' \in B} (s' - \tilde{s}')^2$$

The critic's epistemic uncertainty regarding a particular state is simply the standard deviation of the three predictive heads' predictions. The small predictive networks are a component of the larger ensemble of critics and predictive networks that make up the Predictive Uncertainty Networks.

Note that we intentionally use the same $\xi_i$ as input to both the final head of the critic and as input to the predictive network so that the quality of the two predictions will be tightly coupled. This allows the disagreement among the predictive networks to be highly indicative of the critic's epistemic uncertainty. When initially the critic backbone's latent representations are meaningless, the predictive networks' outputs will be inconsistent with one another and have a large standard deviation. As the critic network better learns to approximate the optimal value manifold, the latent representation will better encode the next state the agent will observe for the final critic layer to predict the next Q value as $Q(s_t, a_t) = r_t + \gamma * Q(s_{t+1}, a_{t+1})$. As the critic improves and the latent representation better encodes $s_{t+1}$, the predictive heads' outputs $s'_{t+1}$ will improve and become more consistent, having a smaller standard deviation among each other. Lastly, we intentionally use an identical architecture for the predictive network as with the critic network to ensure they have the same learning capacity and remain tightly coupled.

### 3.2 Goal Selection

QTA samples previously seen goals from the replay buffer to serve as goal candidates. In order to get a precise estimate of epistemic uncertainty $\epsilon$ for a particular state $s$, we uniformly sample $d$ random

actions from the action space and compute the epistemic uncertainty of a state to be the mean of $d$ uncertainty estimates for that state. More precisely:

$$\epsilon_k = \frac{1}{d} \sum_{j=0}^{d} \frac{1}{G} \sum_{i=0}^{G} P_i(Q_{backbone,i}(s, a_j, g_{desired}))$$

We use the desired goal sampled from the environment for these estimates, though the prediction is learned to be agnostic to the choice of goal. The distribution of estimated epistemic uncertainties is normalized between 0 and 1 before proceeding to goal selection. Each goal is sampled with a probability linearly proportionate to its estimated epistemic uncertainty. This simple linear sampling approach encourages QTA to sample goals with high epistemic uncertainty, allowing the agent to gather maximally informative transitions in the regions of the state space where it is least certain. Given a slope $m$ and y-intercept $b$, a goal candidate with epistemic uncertainty $\epsilon_k$ is selected as the agent's next goal with probability:

$$\frac{max(m\epsilon_k + b, 0)}{\sum_{k=0}^{N} max(m\epsilon_k * b, 0)}$$

Whenever the agent successfully reaches its goal, a new goal is sampled using this same procedure. Estimating the agent's epistemic uncertainty only requires $d$ forward passes through the critic backbone and predictive head, so QTA is able to efficiently consider an arbitrarily large number of goal candidates by utilizing batch operations. In practice, $d$ is small.

### 3.3 PRIORITIZED EXPERIENCE REPLAY

To demonstrate the importance of learning from state transitions with high epistemic uncertainty, we implement a modified version of prioritized experience replay (PER). Our implementation ensures that our agent samples the transitions with greatest epistemic uncertainty when performing gradient updates. We make two key modifications to PER. Firstly, rather than using temporal difference error to establish priorities in the replay buffer, we utilize the estimated epistemic uncertainty according to PUN to provide a more accurate measure of which transitions are most beneficial to sample.

Secondly, once a transition is sampled its goal is probabilistically replaced with the goal of a transition later in its trajectory and its reward is recomputed, as in the standard hindsight experience replay (HER) (Andrychowicz et al., 2017) mechanism. That is to say, that in the sparse reward setting we use the PER mechanism to sample from a HER replay buffer instead of a standard replay buffer. This maintains the benefits of the HER replay buffer, while allowing us to sample the most important transitions through the PER mechanism. Similarly to in Schaul et al. (2016), we sample a transition with priority $p_j$ with probability:

$$\frac{p_j^{\alpha}}{\sum_j p_j^{\alpha}}$$

Schaul et al. (2016) demonstrated that oversampling high-priority transitions introduces a bias that can destabilize gradient updates. In order to anneal the bias created by oversampling transitions with high epistemic uncertainty, we multiply by importance sampling weights:

$$w_j = (\frac{1}{N} \frac{1}{p_j})^{\beta}$$

Where $\beta$ is initialized to a set value and gradually increased throughout training until $\beta = 1$, at which point the bias is fully compensated for.

Finally, it is important that the goals selected by QTA are achievable. To promote achievability, QTA eliminates invalid goal candidates from being selected as the first goal in a rollout through a simple threshold mechanism. Any goal candidates with a mean Q value smaller than $cT$ are eliminated, where $T$ is the maximum length of the rollout. Similar to as in Pitis et al. (2020), this heuristic is not core to the algorithm, and could be replaced with, e.g., a generative model designed to generate achievable goals (Florensa et al., 2018) or a discrminative model to quantify the achievability of goals.

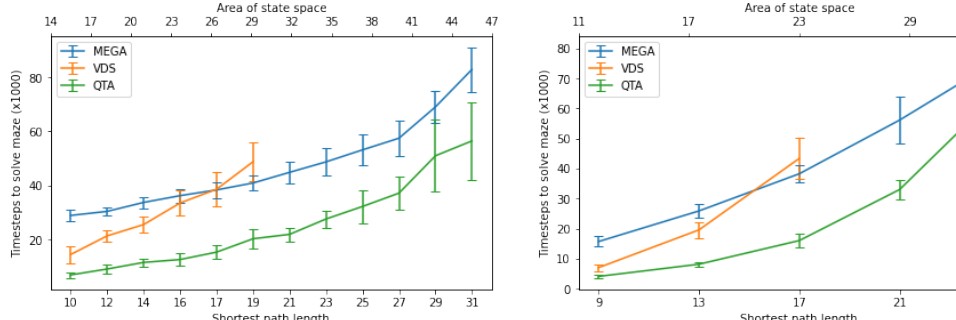

Figure 2: Left: M-maze performance. Right: Square-wave performance. Above we plot the number of timesteps required for various algorithms to attain evaluation performance of 1.0 across 12 seeds on various environments. Performance omitted for when seeds failed to converge for a particular environment. Lower on the graph represents better sample efficiency. Confidence intervals represent one standard deviation.

## 4 RESULTS AND ANALYSIS

**Environments**  We utilize custom a suite of procedurally generated 2D mazes for a dot agent with continuous action and state spaces. In all environments the agent receives a reward of -1 unless it reaches, the goal to within a small radius, in which case it receives a reward of 0. These mazes are carefully designed to have a minimum solving time T such that an optimal agent cannot solve the task in fewer than T timesteps. This reward structure and environment makes it such that the best total for reaching a goal is -1 * (T - 1), which we use in our analysis below to compute the optimal return.

These environments also allow us to evaluate the performance of QTA compared to baseline methods in a series of increasingly challenging environments, where each environment is quantifiably more difficult than the last. Our first set of environments is a set of square-wave mazes, in which the agent must traverse a long path with a single route forward. The square-wave mazes allow us to directly evaluate the most critical skills pertaining to sample efficiency: how quickly an agent is able to explore new regions of the state space and integrate information, while reintegrating information about any regions of the state space that are catastrophically forgotten. Our second set of environments is an M-maze, which features a large, useless dead-end that takes up a large fraction of the state space and is designed to waste that agent's time. While evaluating many of the same challenges, it additionally allows us to evaluate agents' sample efficiency when faced with an artificially enlarged state space.

**Baseline algorithms**  We compare QTA's performance with two competitive curriculum baselines: MEGA (Pitis et al., 2020) and VDS (Zhang et al., 2020). HER was not able to reach the goal in any of our environments within the time allowed. We used the authors' code and hyperparameters for MEGA and VDS, and tried to tune the hyperparameters to our environment. We were unable to improve performance further, likely because both algorithms were originally tuned for very similar 2D navigation environments.

As baselines for comparing PUNs performance, we evaluate QTA with two popular epistemic uncertainty estimation algorithms: Bootstrapped Q Networks (Osband et al., 2016), and Deep Ensembles (Lakshminarayanan et al., 2017). We also evaluate PUN using the discrepancy among Q values as an epistemic uncertainty estimate in order to evaluate the efficacy of predicting next observed states with PUN.

**Main results**  By accurately estimating the epistemic uncertainty of the agent, QTA is able to navigate mazes using significantly fewer timesteps faster than MEGA, as shown in figure 2. MEGA, which ignores the epistemic uncertainty of the model and largely relies on high-level statistics about visited states, did not induce learning as quickly as QTA's methodology of taking into account the epistemic uncertainty. Interestingly, we see that as the time horizon continues to expand to very long horizons and large state spaces, MEGA begins to approach the performance of QTA, which may be due to the predictive network overfitting when predicting the agent's epistemic uncertainties.

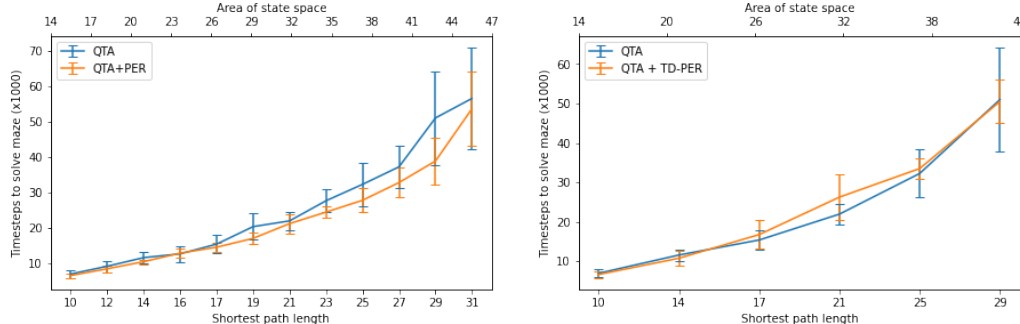

Figure 3: Left: QTA with and without our customized PER. Right: QTA vs standard PER. We plot the number of timesteps needed to attain evaluation performance of 1.0 across 12 seeds in M-mazes of various lengths.

We find that VDS is competitive at first, but quickly becomes less sample efficient and then fails to solve many environments. We suspect that VDS is only competitive at the beginning because VDS has access to privileged information about the entire state space, including states not yet observed.

We also observe that all algorithms perform much worse and scale more poorly on square-wave-shaped environments, despite having much shorter mazes to solve with much smaller state spaces. The greater number of sharp turns required to solve the square-wave mazes hurts sample efficiency, as the optimal value function is harder to approximate.

**Epistemic Uncertainty for Sampling Replay Transitions**   We further demonstrate the importance of estimating the agent's epistemic uncertainty by showing that it can be used as a signal to prioritize certain replay transitions, as shown in figure 3. In the left figure, we show that introducing PER offers some performance benefits with longer mazes before the predictive network begins to overfit. In the right figure. We demonstrate that standard PER with temporal-difference error for prioritization offers no significant performance benefits. Prioritizing based on temporal difference was not helpful to the agent. Prioritizing with epistemic uncertainty accelerated the agent's learning, especially in navigating longer mazes.

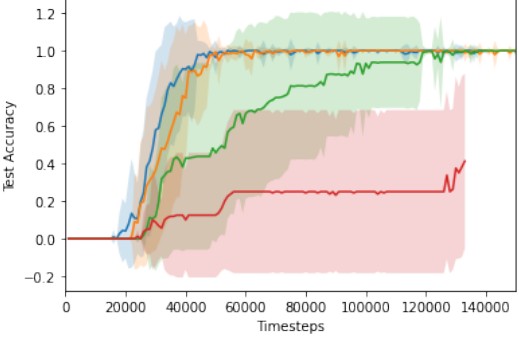

Figure 4: We add gaussian noise to PUN's uncertainty estimates. Blue: no noise. Orange: standard deviation=0.01. Green: standard deviation=0.03. Red: standard deviation=0.05

**Verifying QTA's Approach**   Next, we verify that QTA is forcing the agent to minimize uncertainty and not simply choosing goals that are incrementally further away or following any other strategy. To demonstrate this, as shown in figure 4, we add small amounts of gaussian noise to PUN's uncertainty estimates and find that performance suffers significantly and ultimately collapses with even a tiny amount of added noise. We demonstrate that the performance of QTA hinges on following highly precise estimates of epistemic uncertainty and that QTA selects goals strictly based on epistemic uncertainty.

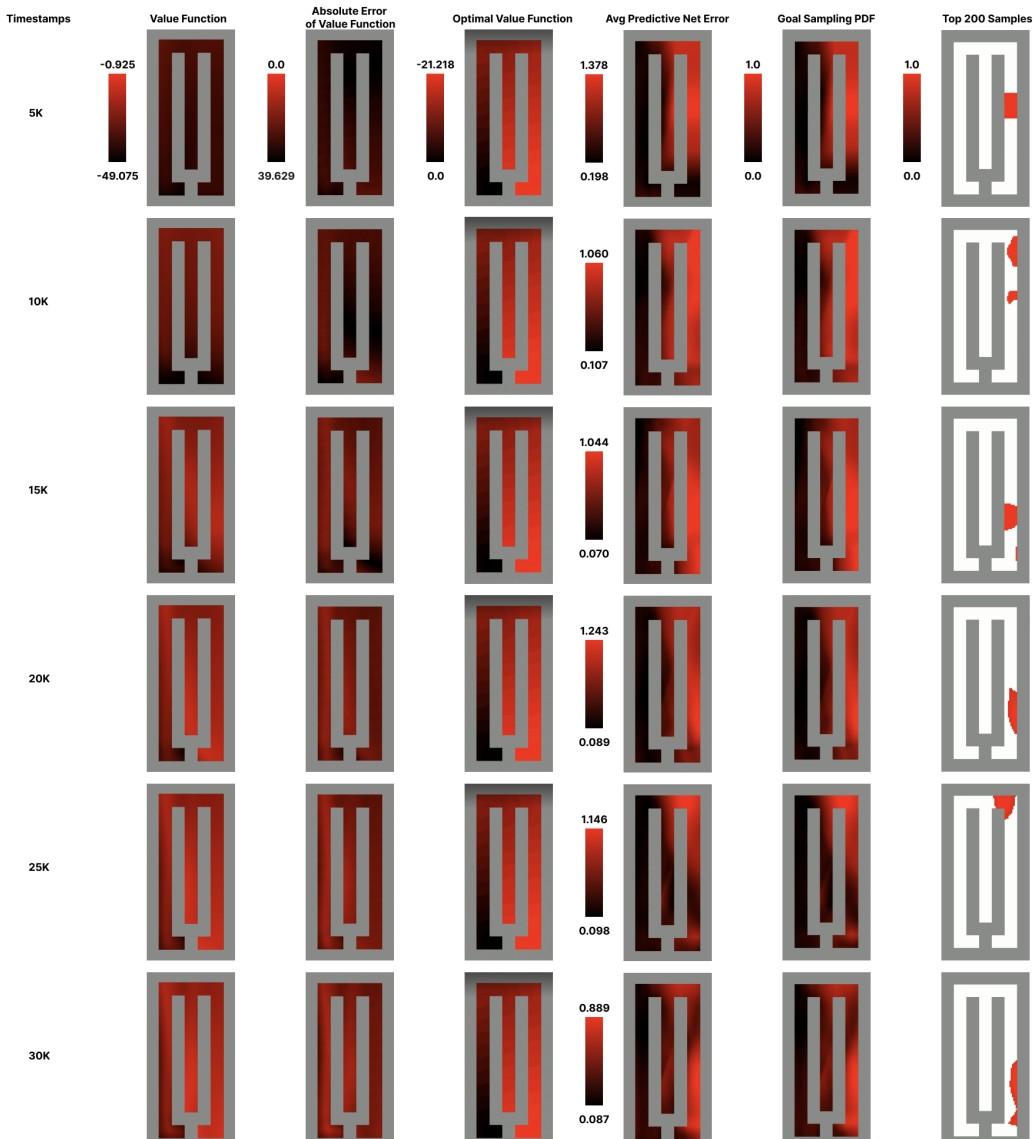

Figure 5: Analysis of the ensemble of critics for one seed on a 12x5-unit M-maze. From left to right, in each column we show: 1) the value function for all states averaged across all critics 2) the absolute value of the error between the average value function and the optimal value function, 3) the optimal value function 4) the average error among the three predictive heads 5) the probability of a state being sampled as the agent's next goal 6) 200 sampled states based on epistemic uncertainty. We emphasize to the reader that across time all color scales are the same across time except for the predictive head error, which changes over time to make the contrast more visible.

**Analysis of critic networks** In figure 5 we visualize the critics of one of our agents learning over time in an M-maze. The agent starts in the bottom-left inlet, as shown in figure 1. We demonstrate that our neural networks are performing as we would expect. Firstly, upon the analysis of the value function, we see that the map generally gets lighter over time, indicating that the estimated Q values increase over time as the agent's competency increases. Naturally, some areas of the value manifold are better learned than others.

Comparing this learned value function with the optimal value function yields the absolute error map shown in the second column. We show that in general the error starts high and uniform, and decreases

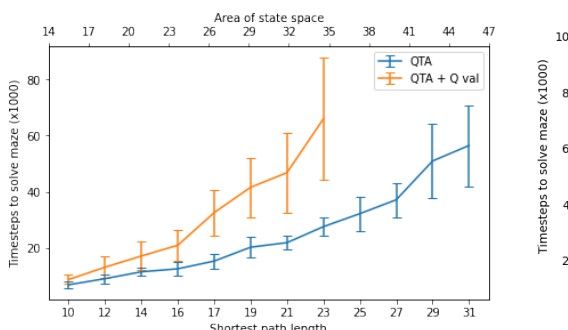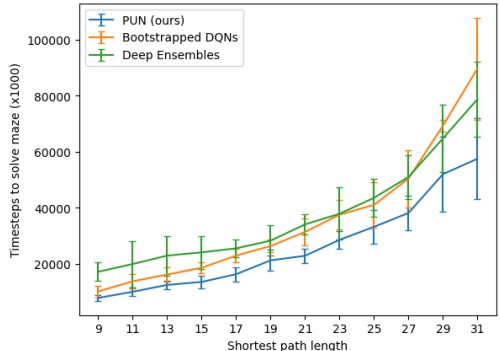

Figure 6: Left: Performance of QTA using PUN uncertainty vs using disagreement among Q values for PUN on various M-mazes. Right: QTA with PUN vs other critic ensemble architectures across M-mazes of varying lengths. All experiments averaged from 12 random seeds. Error bars indicate one standard deviation.

over time. As expected, the error is greater in states further away from the starting point, as those states have been visited less frequently and been sampled fewer times in gradient updates than those closer to the start.

By analyzing the estimated epistemic uncertainty in the fourth column, we gain insight into how the agent selects goals. The fourth column contains a spatial representation of the agent's epistemic uncertainty where brighter colors indicate greater uncertainty. We simultaneously discuss the fifth column, showing the likelihood of any particular point being sampled as a goal for the agent. We see from the average predictive head errors that after only five-thousand iterations, the network already understands and has explored a large region of the maze. We can see that the network least understands the furthest regions of the maze. This is demonstrated by the network sampling goals only in the regions of the maze furthest from the start. As time progresses, the network overall becomes more certain and the magnitude of the average forward error decreases significantly but is punctuated by occasional catastrophic forgetting events. We point out to the reader that the visual scale of the fourth column is unique to each image, as indicated to the left of the maze graphic, in order to make the contrast more visible to the reader. We see that the goal sampling likelihoods in the fifth column track very closely with the predictive head errors in the fourth column, demonstrating that QTA samples goals in the most epistemically uncertain locations for the agent.

In the final column we have a representative sampling of 200 goals, and see that they tend to be targeted toward areas of the maze that are far away from the origin, tracking very closely with the goal sampling likelihoods.

**Q value disagreement** PUN utilizes the disagreement among predictive networks' outputs to estimate epistemic uncertainty. A more intuitive choice might be to use the disagreement among Q values. We experimented with using the disagreement among Q values rather than the predicted next states as in 6, but found it to be an inferior estimator of epistemic uncertainty, leading to dramatically worse performance. By analyzing the sources of noise, we see that we would anticipate the next state prediction to be less noisy and thus more stable than the prediction for the Q value. By the definition of the Q value $Q(s_t) = r_t + \gamma Q(s'_{t+1}, a_{t+1})$, there are three sources of noise $\delta$ in the estimation of the Q value: $\delta_{r_t}$, $\delta_{s'_{t+1}}$, and $\delta_{\gamma Q(s'_{t+1}, a_{t+1})}$. When predicting $s'$, the predictive network's output has only one source of error: $\delta_{s'}$. Naturally, the signal-to-noise ratio will be much higher with the predictive heads' output. A very low signal-to-noise ratio is consistent with the analysis in Zhang et al. (2020). Further, we intuitively anticipate $\tilde{s}'$ to be a good predictor of the quality of the critic's latent representation since the accuracy $\tilde{s}'_{t+1}$ is essential for accurately predicting $Q(s_t, a_t)$.

**Epistemic Uncertainty Estimation Techniques** In figure 6 we compare QTA with PUN to analogous implementations using bootstrapped Q networks and deep ensembles. All are implemented with three critics. With all algorithms we use next state prediction discrepancies as uncertainty estimates. QTA performs best when it uses the epistemic uncertainty estimates from PUN, demonstrating that estimates from PUN are the most accurate of the approaches available.

**Reproducibility Statement**   We take the reproducibility of our work with the utmost seriousness. For this reason we include all of our code, along with instructions for running our code, in the supplementary materials submission. All experiments were run with 12 random seeds and all plots include confidence intervals to make our results clear. We also provide implementation details and hyperparameter settings in the Methods section and Appendix.

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

## A  APPENDIX

### A.1  HYPERPARAMETERS

| Parameter | Value |
|---|---|
| Batch size | 1024 |
| Network hidden size | 256 |
| HER replay k (Andrychowicz et al., 2017) | 4 |
| Discount factor | 0.99 |
| Actor learning rate | 1e-3 |
| Critic Learning Rate | 2e-3 |
| Predictive Network Learning Rate | 5e-3 |
| d (random actions) | 8 (16 for deep ensembles and bootstrapped DQN) |
| Initial random steps | 1e3 |
| $\epsilon$ | 0.2 |
| $p_{randomaction}$ | 0.3 |
| c (threshold) | -1.6 |
| Linear sampling M | 626 |
| Linear Sampling b | -591 |
| $\rho$ | 0.95 |
| PER $\beta_{initial}$ | 0.3 |
| PER $\beta$ steps | 2e5 |
| PER $\alpha$ | 1.0 |

