# OpenReview forum: "Query The Agent: Improving Sample Efficiency Through Epistemic Uncertainty Estimation"
_ICLR.cc/2023/Conference — Submitted to ICLR 2023_

### Official Review · Reviewer_tn6E · 2022-10-24

**Confidence:** 3
**Correctness:** 3
**Technical Novelty And Significance:** 2
**Empirical Novelty And Significance:** 2
**Recommendation:** 5

**Clarity, Quality, Novelty And Reproducibility:**

Overall I think the paper has strong clarity and reproducibility.
Some of the concerns I have are more centered around how incremental the work could be.

**Strength And Weaknesses:**

There are several things to like about this paper:
- The overall quality of writing and polish is pretty high.
- The core QTA algorithm seems reasonable, and some of the pieces of network design/modifcations to ensemble uncertainty to form "PUN" seem useful.
- The experiments and analysis are quite well presented.
- The accompanying code and reproducibility appears to be extremely high.

However there are a few places where the paper could be improved:
- The improvements relative to baselines (MEGA, BootDQN, ...) all seem to be more incremental than transformative. Typically, when people look at these difficult maze-style domains you really care about *scalability* rather than pure performance (since these are just toy problems after all)... in terms of scalability, it does not seem very significant differences as the problems grow.
- Related to the point above, these experiments seem a little "adhoc"... I would have liked to see something like a regret analysis/proof for tabular settings and/or an evaluation on some kind of scalable domain like DeepSea (behaviour suite for reinforcement learning).
- The discussion of related work, while pretty comprehensive, seems quite piecemeal.. I think it would be possible to consolidate and digest some of the related approaches (note that many of these are essentially ensemble-based uncertainty). It would be nice to get some kind of bigger-picture evaluation of what types of uncertainty and techniques really are what make QTA valuable versus these other approaches. Looking at something like "Deep Exploration via Randomized Value Functions" (Osband et al) might be a good overview of the type of approximate posterior sampling that many of these algorithms instantiate..
- Some pieces of the algorithm/design could benefit from being spelled out more explicity. For example, action selection is only described via reference to the DDPG paper... I think this should be made more explicit... what type of exploration policy is used in action selection? I think this is unusual if it relies on the same kind of boltzmann/dithering as DDPG... could not scale to large domains that require deep exploration.



**Summary Of The Paper:**

This paper introduces a new reinforcement learning algorithm: Query the Agent.
This algorithm uses uncertainty estimates derived from an ensemble actor-critic setup to bias replay and learning around more uncertain states and actions.
The authors support their claims mainly through evaluation on an "M-maze" experiment, where QTA outperforms baselines.

**Summary Of The Review:**

This paper presents a new RL algorithm designed to leverage epistemic uncertainty estimates for improved learning.
Overall, I think there are several things to like about the paper, but I'm left with more of a feeling that these improvements are more incremental than transformative.
This could be a reasonable paper in the conference, but I think it could benefit from more clear and decisive results and/or comparisons to existing work.

---

### Official Review · Reviewer_beUS · 2022-10-25

**Confidence:** 3
**Correctness:** 3
**Technical Novelty And Significance:** 2
**Empirical Novelty And Significance:** 2
**Recommendation:** 5

**Clarity, Quality, Novelty And Reproducibility:**

**Clarity**

The paper is well written, and easy to follow with sufficient details described for the method.


**Quality and Novelty**

While some parts of the proposed approach such as the uncertainty estimation method are novel, the overall approach is not novel. The empirical analysis is through and well done but the limited variety of environments considered limits impact.

**Reproducibility**

The authors provide code along with the submission, and most of the method details are described in the paper.

**Strength And Weaknesses:**

**Strengths**

- Leveraging the epistemic uncertainty of the learner to establish a curriculum of goal for training is a neat idea and is a natural way to characterize the learning process - the agent gradually learns to achieve goals that it has high uncertainty over - leading to improved sample efficiency.
- The empirical analysis in the paper is quite thorough and well done.

**Weaknesses**

- In my view the main weakness of the paper is the limited experimental setups considered. The experiments are done only on 2D mazes which do not seem sufficient as evidence for the claims made in the paper.
- Using the uncertainty estimates directly has the potential to suffer from the noisy-TV problem. That is, states of higher-entropy might be selected as goals even though they do not serve as informative goals for the agent.
- The idea of using epistemic uncertainty to guide exploration in reinforcement learning is not particularly novel and has been explored in the past [1, 2]. [2] also addresses the previous point as instead of simply using the uncertainty to weigh the points, they define an acquisition function to quantify the information gain.

[1] - DEUP: Direct Epistemic Uncertainty Prediction

[2] - An Experimental Design Perspective on Model-Based Reinforcement Learning

**Summary Of The Paper:**

In recent years, goal conditioned reinforcement learning approaches have received a lot of attention in the literature. However, curricula for selecting goals during training have predominantly been based on heuristics, which still lead to poor sample-efficiency in realistic scenarios. The paper proposes using curricula based on the agent's epistemic uncertainty. An agent's epistemic uncertainty is it's lack of knowledge. Using this epistemic uncertainty to select goals provides a natural curriculum where the agent gradually acquires knowledge. The authors propose predictive uncertainty networks, a novel uncertainty estimation method for quantifying the agent's epistemic uncertainty. The uncertainty estimates are used as scores in a prioritized replay-based mechanism for making gradient updates. Additionally the replay buffer also involves hindsight experience-like goal selection. Finally the authors implement the proposed algorithm in a procedurally-generated 2D mazes.

**Summary Of The Review:**

in summary,  while the presented approach is interesting and is accompanied by a through and insightful empirical analysis, the significance of the approach is limited by simple environments and limited novelty. Results on more diverse environments, for instance from bsuite [1], would improve the claims of the paper significantly. I encourage the authors to incorporate the feedback during the discussion and rebuttal.

[1] Behaviour Suite for Reinforcement Learning

---

### Official Review · Reviewer_dsJh · 2022-10-27

**Confidence:** 4
**Correctness:** 2
**Technical Novelty And Significance:** 2
**Empirical Novelty And Significance:** 1
**Recommendation:** 3

**Clarity, Quality, Novelty And Reproducibility:**

The paper is well structured but has many typos. The text has to be proofread to be conveniently readable.

I think that to better realize the novelty of this paper, uncertainty based active learning techniques must be also considered, not just the context of reinforcement learning. This is not addressed in the body of the paper.

The experiments can be run, and the results can possibly be reproduced on the same data set. However, because the implementation has so many parameters, and because most of the code is uncommented and hard to navigate, I doubt 'extended reproducibility' that is applying the algorithm to other domains is going to succeed.

**Strength And Weaknesses:**

Strengths: the paper considers application of techniques of adaptive active learning to goal-conditioned reinforcement learning problems. The idea is explained in great detail, compared with other ideas to account for uncertainty in the literature. The supplementary material is provided, with the code to reproduce all experiments in the paper.

Weaknesses: the paper claims that

1) goals should be set with regions with high epistemic uncertainty;
2) states are described by real vectors;
3) the mean of standard deviations of each component of the state vectors is a good measure of epistemic uncertainty.

The first claim seems to be a feasible option, but unsupported. An alternative option might be, for example, setting goals such that paths pass through regions of high uncertainty (rather than end in them). I do not argue that  a different option can be better, I just point that the choice to set goals where uncertainty is high seems to be rather arbirtrary, unless supported at least by some discussions.

The second claim seems to cover many applications of deep reinforcement learning, however it must be stated explicitly. There are models with other state representations.

The third claim is problematic. There is extensive research in quantify sample-based uncertainty (just one citation for example, https://link.springer.com/article/10.1007/s10994-021-06003-9).  Even if  second-order approximation of the distribution is used, the covariance matrix is only poorly characterized by the sum (or the mean) of the diagonal elements.

The ideas behind the algorithm seem to possibly usable, but unsupported theoretically. An alternative would be to support than experimentally. However, the experiments are performed only on a set of relatively small 2-dimensional mazes. One does not really need deep neural networks with complicated architecture to solve those mazes, so while they can be used to confirm that the idea basically works, they are not a sufficient empirical evaluation for the above heuristic claims. One obvious concern, but there are others, is that mean standard deviation of components will probably work in 2 dimensions, but it will have no chance to work in 20 or more. In high dimensions L2 norm between points in the space vanishes to be informative. It is called 'the curse of dimensionality'.

Finally, judging by both the text of the paper and by the code in the supplementary material, the algorithm has dozens of hyperparameters, (probably set GIVEN the data set of mazes, btw). It is not clearly why those values were chosen, and how to chose the best values in general. Without that, the algorithm is not going to be generalizable.



**Summary Of The Paper:**

The paper proposes a new goal selection algorithm for training curricula of goal-oriented reinforcement learning agents. The algorithm, called query the agent, estimates epistemic uncertainty of the agent in the state space and sets goals in areas with higher epistemic uncertainty. The algorithm is evaluated on a set of generated 2-dimensional mazes.

**Summary Of The Review:**

Interesting research direction, however the results as presented are too early for  publication.

---

### Official Review · Reviewer_6NMf · 2022-10-31

**Confidence:** 2
**Correctness:** 4
**Technical Novelty And Significance:** 1
**Empirical Novelty And Significance:** Not applicable
**Recommendation:** 5

**Clarity, Quality, Novelty And Reproducibility:**

The paper is clear and well written., but addresses a limited audience, perhaps no larger than the authors of some of the recent papers it cites.  As for understandability the authors do offer full versions of the code used for the simulations. However in terms of following the paper's argument its explanations are lacking. Save for an example of the value of curriculum learning, the introduction does not give insight into the method, relying on newly coined terms such as PER, without description of what they entail. The reader gets a clue about how the paper uses the term "epistemic uncertainty" when mentioned in the introduction as "encouraging agents to explore regions of the state space the agent least understands" -- but this would apply to any exploration method.  it would be better if critical and new terms were introduced early and defined in a way to make the paper accessible to a wider audience.

Also I suggest the authors number equations in the final version.

**Strength And Weaknesses:**

Granted no familiarity with the few recent papers on which this work depends, my sense is that this is a study of an incremental exploration of a set of proposed modifications to RL critics demonstrated in a toy domain.   It's not clear what the larger relevance and value of this work is.  The paper may be "correct" but the take-aways appear marginal if any.

**Summary Of The Paper:**

"Query The Agent" is an improvement on curriculum learning for deep reinforcement learning. If I understand, to remedy the data inefficiency of RL, this approach applies "curricula" in the form of intermediate learning goals that are informed by the epistemic uncertainty of the agent, e.g. the agent's lack of understanding of parts of the state space.  The agent benefits by exploring (trying to achieve) appropriate goals -- those not to easy or too hard. Uncertainty is represented by running multiple critics.  "Epistemic uncertainty" is estimated by running multiple critics, based on a neural network predictor called a "predictive uncertainty network" and looking at their standard deviation.

The modifications tested in the paper show measurable improvement compared to recent "MEGA" and "VDS" methods.

**Summary Of The Review:**

The high level impression from the paper is of an minor improvement made among several related recent approaches in a simulated environment.  I appreciate the value of simple experiments to illustrate a point, but wonder where the applicability might be to less specific and more realistic problems.  To truly evaluate the paper one needs to understand the related "baseline" work from a few specific papers from the past few years to which the current work is compared.

---

### Decision · Program_Chairs · 2023-01-20

**Decision:**

Reject

**Justification For Why Not Higher Score:**

As detailed in my main meta-review, all reviewers raised concerns about the technical contributions of this work which indicate it doesn't meet the bar for acceptance. As there was no rebuttal, I think it's appropriate to reject.

**Justification For Why Not Lower Score:**

N/A

**Metareview: Summary, Strengths And Weaknesses:**

The authors introduce a new method for curriculum learning via proposing intermediate goals informed by the epistemic uncertainty of the agent, evaluated on procedurally generated 2D mazes.

Reviewers praised the overall nice presentation and clarity of the work, and commended that the code was released along with the submission, thus conferring it greater reproducibility. However, there were some main critiques raised that indicate this work is not suitable for acceptance in its current form. Namely, 1) their particular metric for epistemic uncertainty (standard deviation of components of the state vectors) seems arbitrary and without theoretical support, 2) experiments were only done on 2D mazes , and 3) comparison and discussion of related work could be improved. In the end all reviewers indicated they didn’t think this work was ready for publication just yet.

It’s disappointing that authors didn’t provide any rebuttal or update their paper (as far as I could see), and thus I cannot recommend acceptance, but hope that they find the reviewers’ feedback helpful for future versions of this work.